# Measuring Emotions in the COVID-19 Real World Worry Dataset

**Bennett Kleinberg**[1,2]      **Isabelle van der Vegt**[1]      **Maximilian Mozes**[1,2,3]

[1]Department of Security and Crime Science
[2]Dawes Centre for Future Crime
[3]Department of Computer Science
University College London

{bennett.kleinberg, isabelle.vandervegt, maximilian.mozes}@ucl.ac.uk

## Abstract

The COVID-19 pandemic is having a dramatic impact on societies and economies around the world. With various measures of lockdowns and social distancing in place, it becomes important to understand emotional responses on a large scale. In this paper, we present the first ground truth dataset of emotional responses to COVID-19. We asked participants to indicate their emotions and express these in text. This resulted in the *Real World Worry Dataset* of 5,000 texts (2,500 short + 2,500 long texts). Our analyses suggest that emotional responses correlated with linguistic measures. Topic modeling further revealed that people in the UK worry about their family and the economic situation. Tweet-sized texts functioned as a call for solidarity, while longer texts shed light on worries and concerns. Using predictive modeling approaches, we were able to approximate the emotional responses of participants from text within 14% of their actual value. We encourage others to use the dataset and improve how we can use automated methods to learn about emotional responses and worries about an urgent problem.

## 1   Introduction

The outbreak of the SARS-CoV-2 virus in late 2019 and subsequent evolution of the COVID-19 disease has affected the world on an enormous scale. While hospitals are at the forefront of trying to mitigate the life-threatening consequences of the disease, practically all societal levels are dealing directly or indirectly with an unprecedented situation. Most countries are — at the time of writing this paper — in various stages of a lockdown. Schools and universities are closed or operate online-only, and merely essential shops are kept open.

At the same time, lockdown measures such as social distancing (e.g., keeping a distance of at least 1.5 meters from one another and only socializing with two people at most) might have a direct impact on people's mental health. With an uncertain outlook on the development of the COVID-19 situation and its preventative measures, it is of vital importance to understand how governments, NGOs, and social organizations can help those who are most affected by the situation. That implies, at the first stage, understanding the emotions, worries, and concerns that people have and possible coping strategies. Since a majority of online communication is recorded in the form of text data, measuring the emotions around COVID-19 will be a central part of understanding and addressing the impacts of the COVID-19 situation on people. This is where computational linguistics can play a crucial role.

In this paper, we present and make publicly available a high quality, ground truth text dataset of emotional responses to COVID-19. We report initial findings on linguistic correlates of emotions, topic models, and prediction experiments.

### 1.1   Ground truth emotions datasets

Tasks like emotion detection (Seyeditabari et al., 2018) and sentiment analysis (Liu, 2015) typically rely on labeled data in one of two forms. Either a corpus is annotated on a document-level, where individual documents are judged according to a predefined set of emotions (Strapparava and Mihalcea, 2007; Preoţiuc-Pietro et al., 2016) or individual $n$-grams sourced from a dictionary are categorised or scored with respect to their emotional value (Bradley et al., 1999; Strapparava and Valitutti, 2004). These annotations are done (semi) automatically (e.g., exploiting hashtags such as #happy) (Mohammad and Kiritchenko, 2015; Abdul-Mageed and Ungar, 2017) or manually through third persons (Mohammad and Turney, 2010). While these approaches are common practice and have accelerated the progress that was made in the field, they are limited in that they prop-

agate a *pseudo* ground truth. This is problematic because, as we argue, the core aim of emotion detection is to make an inference about the author's emotional state. The text as the product of an emotional state then functions as a proxy for the latter. For example, rather than wanting to know whether a Tweet is written in a pessimistic tone, we are interested in learning whether the author of the text actually felt pessimistic.

The limitation inherent to third-person annotation, then, is that they might not be adequate measurements of the emotional state of interest. The solution, albeit a costly one, lies in ground truth datasets. Whereas real ground truth would require - in its strictest sense - a random assignment of people to experimental conditions (e.g., one group that is given a positive product experience, and another group with a negative experience), variations that rely on self-reported emotions can also mitigate the problem. A dataset that relies on self-reports is the *International Survey on Emotion Antecedents and Reactions* (ISEAR)[1], which asked participants to recall from memory situations that evoked a set of emotions. The COVID-19 situation is unique and calls for novel datasets that capture people's affective responses to it while it is happening.

## 1.2 Current COVID-19 datasets

Several datasets mapping how the public responds to the pandemic have been made available. For example, tweets relating to the Coronavirus have been collected since March 11, 2020, yielding about 4.4 million tweets a day (Banda et al., 2020). Tweets were collected through the Twitter stream API, using keywords such as 'coronavirus' and 'COVID-19'. Another Twitter dataset of Coronavirus tweets has been collected since January 22, 2020, in several languages, including English, Spanish, and Indonesian (Chen et al., 2020). Further efforts include the ongoing Pandemic Project[2] which has people write about the effect of the coronavirus outbreak on their everyday lives.

## 1.3 The COVID-19 Real World Worry Dataset

This paper reports initial findings for the *Real World Worry Dataset* (RWWD) that captured the emotional responses of UK residents to COVID-19

at a point in time where the impact of the COVID-19 situation affected the lives of all individuals in the UK. The data were collected on the 6th and 7th of April 2020, a time at which the UK was under "lockdown" (news, 2020), and death tolls were increasing. On April 6, 5,373 people in the UK had died of the virus, and 51,608 tested positive (Walker , now). On the day before data collection, the Queen addressed the nation via a television broadcast (Guardian, 2020). Furthermore, it was also announced that Prime Minister Boris Johnson was admitted to intensive care in a hospital for COVID-19 symptoms (Lyons, 2020).

The RWWD is a ground truth dataset that used a direct survey method and obtained written accounts of people alongside data of their felt emotions while writing. As such, the dataset does not rely on third-person annotation but can resort to direct self-reported emotions. We present two versions of RWWD, each consisting of 2,500 English texts representing the participants' genuine emotional responses to Corona situation in the UK: the Long RWWD consists of texts that were open-ended in length and asked the participants to express their feelings as they wish. The Short RWWD asked the same people also to express their feelings in Tweet-sized texts. The latter was chosen to facilitate the use of this dataset for Twitter data research.

The dataset is publicly available.[3]

## 2 Data

We collected the data of $n = 2500$ participants (94.46% native English speakers) via the crowd-sourcing platform Prolific[4]. Every participant provided consent in line with the local IRB. The sample requirements were that the participants were resident in the UK and a Twitter user. In the data collection task, all participants were asked to indicate how they felt about the current COVID-19 situation using 9-point scales (1 = not at all, 5 = moderately, 9 = very much). Specifically, each participant rated how worried they were about the Corona/COVID-19 situation and how much anger, anxiety, desire, disgust, fear, happiness, relaxation, and sadness (Harmon-Jones et al., 2016) they felt about their situation at this moment. They also had to choose which of the eight emotions (ex-

---

[1] https://www.unige.ch/cisa/research/materials-and-online-research/research-material/
[2] https://utpsyc.org/covid19/index.html

[3] Data: https://github.com/ben-aaron188/covid19worry and https://osf.io/awy7r/
[4] https://www.prolific.co/

cept worry) best represented their feeling at this moment.

All participants were then asked to write two texts. First, we instructed them to "*write in a few sentences how you feel about the Corona situation at this very moment. This text should express your feelings at this moment*" (min. 500 characters). The second part asked them to express their feelings in Tweet form (max. 240 characters) with otherwise identical instructions. Finally, the participants indicated on a 9-point scale how well they felt they could express their feelings (in general/in the long text/in the Tweet-length text) and how often they used Twitter (from 1=never, 5=every month, 9=every day) and whether English was their native language. The overall corpus size of the dataset was 2500 long texts (320,372 tokens) and 2500 short texts (69,171 tokens). In long and short texts, only 6 and 17 emoticons (e.g. ":(", "<3") were found, respectively. Because of the low frequency of emoticons, these were not focused on in our analysis.

## 2.1 Excerpts

Below are two excerpts from the dataset:

**Long text:** *I am 6 months pregnant, so I feel worried about the impact that getting the virus would have on me and the baby. My husband also has asthma so that is a concern too. I am worried about the impact that the lockdown will have on my ability to access the healthcare I will need when having the baby, and also about the exposure to the virus [...] There is just so much uncertainty about the future and what the coming weeks and months will hold for me and the people I care about.*

**Tweet-sized text:** *Proud of our NHS and keyworkers who are working on the frontline at the moment. I'm optimistic about the future, IF EVERYONE FOLLOWS THE RULES. We need to unite as a country, by social distancing and stay in.*

## 2.2 Descriptive statistics

We excluded nine participants who padded the long text with punctuation or letter repetitions. The dominant feelings of participants were anxiety/worry, sadness, and fear (see Table 1)[5]. For all emotions,

the participants' self-rating ranged across the whole spectrum (from "not at all" to "very much"). The final sample consisted to 65.15% of females[6] with an overall mean age of 33.84 years ($SD = 22.04$).

The participants' self-reported ability to express their feelings, in general, was $M = 6.88$ ($SD = 1.69$). When specified for both types of texts separately, we find that the ability to express themselves in the long text ($M = 7.12$, $SD = 1.78$) was higher than that for short texts ($M = 5.91$, $SD = 2.12$), Bayes factor $> 1e + 96$.

The participants reported to use Twitter almost weekly ($M = 6.26$, $SD = 2.80$), tweeted themselves rarely to once per month ($M = 3.67$, $SD = 2.52$), and actively participated in conversations in a similar frequency ($M = 3.41$, $SD = 2.40$). Our participants were thus familiar with Twitter as a platform but not overly active in tweeting themselves.

| Variable | Mean | SD |
|---|---|---|
| *Corpus descriptives* | | |
| Tokens (long text) | 127.75 | 39.67 |
| Tokens (short text) | 27.70 | 15.98 |
| Types (long text) | 82.69 | 18.24 |
| Types (short text) | 23.50 | 12.21 |
| TTR (long text) | 0.66 | 0.06 |
| TTR (short text) | 0.88 | 0.09 |
| Chars. (long text) | 632.54 | 197.75 |
| Chars. (short text) | 137.21 | 78.40 |
| | | |
| *Emotions* | | |
| Worry | 6.55[a] | 1.76 |
| Anger[1] (4.33%) | 3.91[b] | 2.24 |
| Anxiety (55.36%) | 6.49[a] | 2.28 |
| Desire (1.09%) | 2.97[b] | 2.04 |
| Disgust (0.69%) | 3.23[b] | 2.13 |
| Fear (9.22%) | 5.67[a] | 2.27 |
| Happiness (1.58%) | 3.62[b] | 1.89 |
| Relaxation (13.38%) | 3.95[b] | 2.13 |
| Sadness (14.36%) | 5.59[a] | 2.31 |

Table 1: Descriptive statistics of text data and emotion ratings. [1]brackets indicate how often the emotion was chosen as the best fit for the current feeling about COVID-19. [a]the value is larger than the neutral midpoint with Bayes factors $> 1e + 32$. [b]the value is smaller than the neutral midpoint with BF $> 1e + 115$. TTR = type-token ratio.

[5]For correlations among the emotions, see the online supplement

[6]For an analysis of gender differences using this dataset, see van der Vegt and Kleinberg (2020).

# 3 Findings and experiments

## 3.1 Correlations of emotions with LIWC categories

We correlated the self-reported emotions to matching categories of the LIWC2015 lexicon (Pennebaker et al., 2015). The overall matching rate was high (92.36% and 90.11% for short and long texts, respectively). Across all correlations, we see that the extent to which the linguistic variables explain variance in the emotion values (indicated by the $R^2$) is larger in long texts than in Tweet-sized short texts (see Table 2). There are significant positive correlations for all affective LIWC variables with their corresponding self-reported emotions (i.e., higher LIWC scores accompanied higher emotion scores, and vice versa). These correlations imply that the linguistic variables explain up to 10% and 3% of the variance in the emotion scores for long and short texts, respectively.

The LIWC also contains categories intended to capture areas that concern people (not necessarily in a negative sense), which we correlated to the self-reported worry score. Positive (negative) correlations would suggest that the higher (lower) the worry score of the participants, the larger their score on the respective LIWC category. We found no correlation between the categories "work", "money" and "death" suggesting that the worry people reported was not associated with these categories. Significant positive correlations emerged for long texts for "family" and "friend": the more people were worried, the more they spoke about family and — to a lesser degree — friends.

## 3.2 Topic models of people's worries

We constructed topic models for both the long and short texts separately using the stm package in R (Roberts et al., 2014a). The text data were lowercased, punctuation, stopwords and numbers were removed, and all words were stemmed. For the long texts, we chose a topic model with 20 topics as determined by semantic coherence and exclusivity values for the model (Mimno et al., 2011; Roberts et al., 2014b,a). Table 3 shows the five most prevalent topics with ten associated frequent terms for each topic (see online supplement for all 20 topics). The most prevalent topic seems to relate to following the rules related to the lockdown. In contrast, the second most prevalent topic appears to relate to worries about employment and the economy. For the Tweet-sized texts, we selected a model with 15 topics. The most common topic bears a resemblance to the government slogan "Stay at home, protect the NHS, save lives." The second most prevalent topic seems to relate to calls for others to adhere to social distancing rules.

## 3.3 Predicting emotions about COVID-19

It is worth noting that the current literature on automatic emotion detection mainly casts this problem as a classification task, where words or documents are classified into emotional categories (Buechel and Hahn, 2016; Demszky et al., 2020). Our fine-grained annotations allow for estimating emotional values on a continuous scale. Previous works on emotion regression utilise supervised models such as linear regression for this task (Preoţiuc-Pietro et al., 2016), and more recent efforts employ neural network-based methods (Wang et al., 2016; Zhu et al., 2019). However, the latter typically require larger amounts of annotated data, and are hence less applicable to our collected dataset.

We, therefore, use linear regression models to predict the reported emotional values (i.e., anxiety, fear, sadness, worry) based on text properties. Specifically, we applied regularised ridge regression models[7] using TFIDF and part-of-speech (POS) features extracted from long and short texts separately. TFIDF features were computed based on the 1000 most frequent words in the vocabularies of each corpus; POS features were extracted using a predefined scheme of 53 POS tags in *spaCy*[8].

We process the resulting feature representations using principal component analysis and assess the performances using the mean absolute error (MAE) and the coefficient of determination $R^2$. Each experiment is conducted using five-fold cross-validation, and the arithmetic means of all five folds are reported as the final performance results.

Table 4 shows the performance results in both long and short texts. We observe MAEs ranging between 1.26 (worry with TFIDF) and 1.88 (sadness with POS) for the long texts, and between 1.37 (worry with POS) and 1.91 (sadness with POS) for the short texts. We furthermore observe that the models perform best in predicting the worry scores for both long and short texts. The models explain up to 16% of the variance for the emotional response variables on the long texts, but only up to

---

[7]We used the *scikit-learn* python library (Pedregosa et al., 2011).

[8]https://spacy.io

| Correlates | Long texts | Short texts |
|---|---|---|
| *Affective processes* | | |
| Anger - LIWC "anger" | 0.28 [0.23; 0.32] (7.56%) | 0.09 [0.04; 0.15] (0.88%) |
| Sadness - LIWC "sad" | 0.21 [0.16; 0.26] (4.35%) | 0.13 [0.07; 0.18] (1.58%) |
| Anxiety - LIWC "anx" | 0.33 [0.28; 0.37] (10.63%) | 0.18 [0.13; 0.23] (3.38%) |
| Worry - LIWC "anx" | 0.30 [0.26; 0.35] (9.27%) | 0.18 [0.13; 0.23] (3.30%) |
| Happiness - LIWC "posemo" | 0.22 [0.17; 0.26] (4.64%) | 0.13 [0.07; 0.18] (1.56%) |
| | | |
| *Concern sub-categories* | | |
| Worry - LIWC "work" | -0.03 [-0.08; 0.02] (0.01%) | -0.03 [-0.08; 0.02] (0.10%) |
| Worry - LIWC "money" | 0.00 [-0.05; 0.05] (0.00%) | -0.01 [-0.06; 0.04] (0.00%) |
| Worry - LIWC "death" | 0.05 [-0.01; 0.10] (0.26%) | 0.05 [0.00; 0.10] (0.29%) |
| Worry - LIWC "family" | 0.18 [0.13; 0.23] (3.12%) | 0.06 [0.01; 0.11] (0.40%) |
| Worry - LIWC "friend" | 0.07 [0.01; 0.12] (0.42%) | -0.01 [-0.06; 0.05] (0.00%) |

Table 2: Correlations (Pearson's $r$, 99% CI, $R$-squared in %) between LIWC variables and emotions.

| Docs | Terms |
|---|---|
| *Long texts* | |
| 9.52 | people, take, think, rule, stay, serious, follow, virus, mani, will |
| 8.35 | will, worri, job, long, also, economy, concern, impact, famili, situat |
| 7.59 | feel, time, situat, relax, quit, moment, sad, thing, like, also |
| 6.87 | feel, will, anxious, know, also, famili, worri, friend, like, sad |
| 5.69 | work, home, worri, famili, friend, abl, time, miss, school, children |
| | |
| *Short texts* | |
| 10.70 | stay, home, safe, live, pleas, insid, save, protect, nhs, everyone |
| 8.27 | people, need, rule, dont, stop, selfish, social, die, distance, spread |
| 7.96 | get, can, just, back, wish, normal, listen, lockdown, follow, sooner |
| 7.34 | famili, anxious, worri, scare, friend, see, want, miss, concern, covid |
| 6.81 | feel, situat, current, anxious, frustrat, help, also, away, may, extrem |

Table 3: The five most prevalent topics for long and short texts.

1% on Tweet-sized texts.

| Model | Long | | Short | |
|---|---|---|---|---|
| | MAE | $R^2$ | MAE | $R^2$ |
| Anxiety - TFIDF | 1.65 | 0.16 | 1.82 | -0.01 |
| Anxiety - POS | 1.79 | 0.04 | 1.84 | 0.00 |
| Fear - TFIDF | 1.71 | 0.15 | 1.85 | 0.00 |
| Fear - POS | 1.83 | 0.05 | 1.87 | 0.01 |
| Sadness - TFIDF | 1.75 | 0.12 | 1.90 | -0.02 |
| Sadness - POS | 1.88 | 0.02 | 1.91 | -0.01 |
| Worry - TFIDF | 1.26 | 0.16 | 1.38 | -0.03 |
| Worry - POS | 1.35 | 0.03 | 1.37 | 0.01 |

Table 4: Results for regression modeling for long and short texts.

## 4 Discussion

This paper introduced a ground truth dataset of emotional responses in the UK to the Corona pandemic. We reported initial findings on the linguistic correlates of emotional states, used topic modeling to understand what people in the UK are concerned about, and ran prediction experiments to infer emotional states from text using machine learning. These analyses provided several core findings: (1) Some emotional states correlated with word lists made to measure these constructs, (2) longer texts were more useful to identify patterns in language that relate to emotions than shorter texts, (3) Tweet-sized texts served as a means to call for solidarity during lockdown measures while longer

texts gave insights to people's worries, and (4) preliminary regression experiments indicate that we can infer from the texts the emotional responses with an absolute error of 1.26 on a 9-point scale (14%).

## 4.1 Linguistic correlates of emotions and worries

Emotional reactions to the Coronavirus were obtained through self-reported scores. When we used psycholinguistic word lists that measure these emotions, we found weak positive correlations. The lexicon-approach was best at measuring anger, anxiety, and worry and did so better for longer texts than for Tweet-sized texts. That difference is not surprising given that the LIWC was not constructed for micro-blogging and very short documents. In behavioral and cognitive research, small effects (here: a maximum of 10.63% of explained variance) are the rule rather than the exception (Gelman, 2017; Yarkoni and Westfall, 2017). It is essential, however, to interpret them as such: if 10% of the variance in the anxiety score are explained through a linguistic measurement, 90% are not. An explanation for the imperfect correlations - aside from random measurement error - might lie in the inadequate expression of someone's felt emotion in the form of written text. The latter is partly corroborated by even smaller effects for shorter texts, which may have been too short to allow for the expression of one's emotion.

It is also important to look at the overlap in emotions. Correlational follow-up analysis (see online supplement) among the self-reported emotions showed high correlations of worry with fear ($r = 0.70$) and anxiety ($r = 0.66$) suggesting that these are not clearly separate constructs in our dataset. Other high correlations were evident between anger and disgust ($r = 0.67$), fear and anxiety ($r = 0.78$), and happiness and relaxation ($r = 0.68$). Although the chosen emotions (with our addition of "worry") were adopted from previous work (Harmon-Jones et al., 2016), it merits attention in future work to disentangle the emotions and assess, for example, common ngrams per cluster of emotions (e.g. as in Demszky et al., 2020).

## 4.2 Topics of people's worries

Prevalent topics in our corpus showed that people worry about their jobs and the economy, as well as their friends and family - the latter of which is also corroborated by the LIWC analysis. For example, people discussed the potential impact of the situation on their family, as well as their children missing school. Participants also discussed the lockdown and social distancing measures. In the Tweet-sized texts, in particular, people encouraged others to stay at home and adhere to lockdown rules in order to slow the spread of the virus, save lives and/or protect the NHS. Thus, people used the shorter texts as a means to call for solidarity, while longer texts offered insights into their actual worries (for recent work on gender differences, see van der Vegt and Kleinberg, 2020).

While there are various ways to select the ideal number of topics, we have relied on assessing the semantic coherence of topics and exclusivity of topic words. Since there does not seem to be a consensus on the best practice for selecting topic numbers, we encourage others to examine different approaches or models with varying numbers of topics.

## 4.3 Predicting emotional responses

Prediction experiments revealed that ridge regression models can be used to approximate emotional responses to COVID-19 based on encoding of the textual features extracted from the participants' statements. Similar to the correlational and topic modeling findings, there is a stark difference between the long and short texts: the regression models are more accurate and explain more variance for longer than for shorter texts. Additional experiments are required to investigate further the expressiveness of the collected textual statements for the prediction of emotional values. The best predictions were obtained for the reported worry score (MAE = 1.26, MAPE = 14.00%). An explanation why worry was the easiest to predict could be that it was the highest reported emotion overall with the lowest standard deviation, thus potentially biasing the model. More fine-grained prediction analyses out of the scope of this initial paper could further examine this.

## 4.4 Suggestions for future research

The current analysis leaves several research questions untouched. First, to mitigate the limitations of lexicon-approaches, future work on inferring emotions around COVID-19 could expand on the prediction approach (e.g., using different feature sets and models). Carefully validated models could help to provide the basis for large scale, real-time measurements of emotional responses. Of particu-

lar importance is a solution to the problem hinted at in the current paper: the shorter, Tweet-sized texts contained much less information, had a different function, and were less suitable for predictive modeling. However, it must be noted that the experimental setup of this study did not fully mimic a 'natural' Twitter experience. Whether the results are generalisable to actual Twitter data is an important empirical question for follow-up work. Nevertheless, with much of today's stream of text data coming in the form of (very) short messages, it is important to understand the limitations of using that kind of data and worthwhile examining how we can better make inferences from that information.

Second, with a lot of research attention paid to readily available Twitter data, we hope that future studies also focus on non-Twitter data to capture emotional responses of those who are underrepresented (or non-represented) on social media but are at heightened risk.

Third, future research may focus on manually annotating topics to more precisely map out what people worry about with regards to COVID-19. Several raters could assess frequent terms for each topic, then assign a label. Then through discussion or majority votes, final topic labels can be assigned to obtain a model of COVID-19 real-world worries.

Fourth, future efforts may aim for sampling over a longer period to capture how emotional responses develop over time. Ideally, using high-frequency sampling (e.g., daily for several months), future work could account for the large number of events that may affect emotions.

Lastly, it is worthwhile to utilise other approaches to measuring psychological constructs in text. Although the rate of out-of-vocabulary terms for the LIWC in our data was low, other dictionaries may be able to capture other relevant constructs. For instance, the tool Empath (Fast et al., 2016) could help measure emotions not available in the LIWC (e.g., nervousness and optimism). We hope that future work will use the current dataset (and extensions thereof) to go further so we can better understand emotional responses in the real world.

## 5 Conclusions

This paper introduced the first ground truth dataset of emotional responses to COVID-19 in text form. Our findings highlight the potential of inferring concerns and worries from text data but also show some of the pitfalls, in particular, when using concise texts as data. We encourage the research community to use the dataset so we can better understand the impact of the pandemic on people's lives.

## Acknowledgments

This research was supported by the Dawes Centre for Future Crime at UCL.

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
