# OpenReview forum: "Measuring Emotions in the COVID-19 Real World Worry Dataset"
_aclweb.org/ACL/2020/Workshop/NLP-COVID — NLP-COVID-2020_

### Official Review · AnonReviewer1 · 2020-04-28
**Useful dataset**

**Rating:** 8
**Confidence:** 4

**Review:**

Summary: The paper introduces a self-reported dataset of 2500 short and long (each) texts about emotions of people during the COVID-19 pandemic. The paper then analyses the dataset in terms of LIWC properties, output of a topic model (most probable topics) and a linear regression model to predict emotion based on TFIDF and POS-based features. The dataset will certainly be useful. The fact that it is self-reported and has both short and long texts are noteworthy.

Suggestions:
1) Short and long texts by the same participant do not necessarily have to be 'parallel'/'analogous', it seems. If this is indeed the case, I would suggest mentioning so.

2) It would be good to know the reason behind picking worry as the key emotion. (In other words, the choice of calling this a 'worry dataset' and not an 'emotion dataset' is not clear). The question asked to the participants (when they draft their text) does not mention 'worry' explicitly. They are asked to rate how worried they feel. However, in addition, the participants are also asked to record other emotions.

3a) Section 2.2: The description accompanying worry clubs it with 'anxiety'. However, table 1 shows that only 55% participants reported anxiety.

3b) Please elaborate "The participants’ self-reported ability to express their feelings in the long text". Was it a part of the form?

4) It is not clear if the number of tokens is the same as vocabulary. It would be useful to know the vocabulary sizes of the datasets.

5) The github repo also includes LIWC statistics of the texts. This could also potentially be useful.

6) There is a low correlation between the 'concerns' (work, money, death?) and worry. In contrast, the top topics from the model include job and family. Is it surprising?

7) It would be useful to add statistics on how frequently the participants reported using Twitter. This would be helpful to understand the quality of the short text.

8) Observation: The classifier is not state-of-the-art. It would be useful to add citations to papers which use linear regression for emotion analysis.

9) Was the linear regression model also trained using spaCy?

10) Is the low MAE for worry related to the fact that worry was central to the annotation? Could there have been a bias in the annotations? The stdev for worry is also the lowest (as shown in Table 1).

The dataset would certainly be useful for future work. The analysis of the dataset (LIWC, topic models) is very interesting. The paper is easy to follow as well.

---

> ### Author Response · Authors · 2020-05-01
> **Response to reviewer**
>
> Thanks for the encouraging and helpful review.
>
> Please find our response re. your original points in order below:
>
> 1. The participants received the same instructions (“to write a text that expresses their feelings about COVID-19”) for both types of text. If they simply shortened the longer text for the Tweet-size version, we would have expected to see the same kind of topics. We will add a clarification about this in the revision.
> 2. We agree that the focus on “worry” does not fully align with the instructions the participants received. While worries emerged as a general issue, the focus of calling the dataset “emotion dataset” is more adequate indeed. We will take this into account for the revision (as this was also raised by reviewer 2) and also change the name of the dataset to more adequately reflect the focus on emotions.
> 3a) We will adjust the focus to “emotions” more generally. We included worry to be judged as participants might have a broader idea of worries compared to anxiety. However, these two are likely highly correlated (we will add this analysis). The revision will clarify this.
> 3b) We add the data of self-reported ability to section 2.3 Descriptive statistics
> 4. Agreed - we will add voc. size, tokens, and TTR to the corpus descriptives.
> 5. -
> 6. We should have clarified this better. The “concerns” from the LIWC do not necessarily relate to negatively connotated concerns but more broadly “areas that concern people” rather than something they worry about. This could be why the correlation is low. That these areas are also emerging from the topics is interesting but needs clarification. We include this in the revision.
> 7. Yes, fully agreed. We add these statistics to section 2.3 Descriptive statistics.
> 8. Agreed. We will add references here and explore better models. The revision will include these.
> 9. We used sklearn and will clarify.
> 10. We will clarify (see also point 2 above) in the revision better that the core focus was on emotions. Another explanation could be that it naturally was so ingrained in the texts (i.e. people were predominantly worried), that the model was biased towards predicting high worry values. We will explore this further and add insights to the revision.

---

### Official Review · AnonReviewer3 · 2020-04-30
**Good first steps toward understanding the UK public's emotional response toward covid19**

**Rating:** 8
**Confidence:** 4

**Review:**

The focus of this manuscript is to 1) describe a corpus of long and short lengthened emotional responses to the covid19 pandemic by the public, 2) identify meaningful associations between linguistic measures and emotional responses, and 3) develop prediction models for automatically classifying unseen texts according to their emotional response categories.

Quality: The usage of LIWC lexicon to identify topical/linguistic information is a good start. I'm interested in how differently tools like empath (https://github.com/Ejhfast/empath-client) would perform in identifying pre-configured and new topics on this corpus and what additional insights could be drawn from it.

The correlation values appear low, but I'm wondering if that's due to out of vocabulary terms. Perhaps, a quick manual review of LIWC term coverage and a lexicon enrichment might help identify a stronger signal.

Some emotions seem a little hard to tease out or have close relationships e.g., worry, anxiety, fear, etc. It would be interesting to understand commonalities and distinguishing characteristics in terms of linguistic measures of related/close categories.

Clarity: A few points should be clarified or explained: 1) did you collect any additional meta-data about the participants e.g., sex, gender, age, race, professions (essential vs. non-essential workers), etc. that could be useful to contextualize or identify particular worries among groups? 2) did these "texts" also include emoticons that could be used to convey emotional response and topical information?

It would also be interesting to sample more than 2 days. I wonder how the topics will shift as the pandemic unfolds.

Also, I would recommend revisiting the corpus title as its very UK-centric and covers a broad range of emotions. What about "UK COVID19 Public Emotional Response dataset"?

Originality: The coupling of the open survey with traditional linguistic and topic modeling approaches for examining the global threat has some originality. The predictive model serves as an initial baseline. It would be interesting to evaluate other traditional machine learning classifiers to establish reasonable baseline performance.

Significance: The topic is certainly significant and a nice first attempt at obtaining self-reported emotional concerns of the public. This is also a great tool that if deployed more broadly could capture insights across regions, countries, and continents.

Thank you for making this invaluable resource publicly available to researchers!

---

> ### Author Response · Authors · 2020-05-01
> **Response to reviewer**
>
>
> Thanks for the review. The points raised will help us improve the paper for the revision and are much appreciated.
>
> Please find our response re. your original points below:
>
> Re. Empath: great suggestions. We will look into this and add findings to the revision. Thanks a lot for pointing us to this fantastic tool.
> Re. LIWC dictionary coverage: we had that idea too initially but the coverage was relatively high (92.36% and 90.11% for short and long) which we saw as sufficient evidence that OOV terms are not the issue. The additional Empath analysis we will conduct can hopefully assist us in further understanding this.
> Re. emotion overlap: agreed. While these were taken from validated emotion questionnaires (with our addition of “worry”), we think correlational analyses will shed light on potential overlap. We add these to the revision.
> Re. meta data: yes, we have these data and will add them to the demographic statistics under section 2.3 Descriptive statistics
> Re. emoticons: good point - we will check how many there were (if any) and look at what they can tell us. We add information on this to the revision.
> Re. multiple days of sampling: agreed. We were debating this and think it’s something for future work. A complication is that so many events might affect the public’s response that ideally, we’d need high-frequency sampling. That is currently out of the scope of our funding unfortunately.
> Re. dataset name: agreed - we will amend the name to focus more on emotions. The revision will clarify this.
> Re. prediction models: we will explore more models and will add these to the revision (see also Reviewer 1’s point #8).

---

### Official Review · AnonReviewer2 · 2020-05-04
**Valuable resource**

**Rating:** 8
**Confidence:** 5

**Review:**

The paper describes a novel dataset containing answers to a survey that asked participants to describe in short and long text their emotional states and rate their levels of  anger, anxiety, desire, disgust, fear, happiness, relaxation, and sadness. The dataset includes responses from 2500 UK residents collected during a critical period of the cover-19 pandemic.

The paper also investigated the predictive performance of lexicon-based and supervised classifiers. Preliminary analyses of the data showed significant correlations between the respondents emotional ratings and the scores inferred by LIWC for the same emotions. However, these correlations are much higher for longer texts, which is not surprising since LIWC was not created for microblog style content and longer text has more words that can match with the lexicon. Similar trends were observed with a supervised model trained to predict the emotional ratings given the text answers. Finally, topic analysis of the data showed that the main topics expressed in shorter text are different than in longer text.

The paper is well written, well scoped and well executed. The dataset is very interesting and it will be useful for the NLP community not only from the standpoint of understanding how people respond to global crises but also to better understand how the characteristics of social media might influence what kinds of information people decide to share. The cautionary tale about using Twitter to study the public's reactions to these kinds of events is worth a closer look. I am not sure how much we can extrapolate from this, given that in real life people can post more than one tweet about a subject, and they do so spontaneously which is different from being asked to post about something specific on a survey.  Some "experimenter effect” might be at play here.

---

> ### Author Response · Authors · 2020-05-04
> **Response to reviewer**
>
> Thanks for the positive and helpful review.
>
> We appreciate your comments and will incorporate the points raised. We fully agree that the Tweet-style texts elicited in the study do not fully mimic a ‘natural’ setting in which Twitter users may write about their emotions and COVID-19. Indeed, on Twitter people may write multiple Tweets about a subject or reply to other users. Therefore, we have noted this as a limitation in our revised paper.

---

### Decision · Program_Chairs · 2020-05-13

**Decision:**

Accept

**Comment:**

Thank you for your submission!

We are very pleased to accept your paper for inclusion in the proceedings.

Both first submission and first acceptance to the Workshop!

Looking forward to having it presented. (Details to follow.)